# Band of mothers: Childbirth as a female bonding experience

**Tara Tasuji****¹\*, Elaine Reese²\*, Valerie van Mulukom¹,³, Harvey Whitehouse¹**

**1** Centre for the Study of Social Cohesion, University of Oxford, Oxford, United Kingdom, **2** Department of Psychology, University of Otago, Dunedin, New Zealand, **3** Centre for Trust, Peace and Social Relations, Coventry University, Coventry, United Kingdom

\* tara.tasuji@gmail.com (TT); ereese@psy.otago.ac.nz (ER)

**Data Availability Statement:** All relevant data are available from the Open Science Framework (DOI: 10.17605/OSF.IO/3J98K).

**Funding:** This work was supported by an Advanced Grant entitled 'Ritual Modes: Divergent Modes of

## Abstract

Does the experience of childbirth create social bonds among first-time mothers? Previous research suggests that sharing emotionally intense or painful experiences with others leads to "identity fusion," a visceral feeling of oneness with a group that predicts strong forms of prosocial action and self-sacrifice for other group members. This study compared identity fusion with other mothers during pregnancy versus after childbirth in a sample of 164 U.S. women. Eighty-nine mothers in our sample were pregnant with their firstborn, and 75 mothers had given birth to their firstborn up to 6 months prior to the time of data collection. Results demonstrated that identity fusion with other mothers was higher for postpartum mothers than for antenatal mothers. As predicted, among postpartum mothers, those who thought that their childbirth was more painful than a typical childbirth experience reported greater identity fusion with mothers who reported having had a very difficult birth. Postpartum mothers' ruminative thought about the birth mediated the association between level of dysphoria and identity fusion, and identity fusion moderated the association between postpartum mothers' ruminative and reflective thought about the birth and their posttraumatic growth in complex ways. These findings provide evidence that perceived sharedness of the childbirth experience and thoughts about the birth are important to the process of identity fusion with other mothers, and highlight the importance of post-event processing for psychological health.

## Introduction

A minority of mothers experience real or threatened injury or death to themselves or their child during the labor and delivery process. Birthing mothers who experience distress and powerlessness often feel that hospital staff dismiss their experience as normal and unexceptional [1]. A meta-analysis of 28 studies found evidence of posttraumatic stress disorder (PTSD) in a small but significant proportion of mothers in the postpartum period. On average, prevalence of PTSD with respect to childbirth is 4.0% in low-risk or general community samples, but 18.5% in groups of women who are at high risk of physical or mental health complications (e.g., women who experience an emergency caesarean section or have a severe fear of

Ritual, Social Cohesion, Prosociality, and Conflict'
(agreement no. 694986) from the European
Research Council (ERC) under the European
Union's Horizon 2020 Research and Innovation
Programme awarded to HW. The funders had no
role in study design, data collection and analysis,
decision to publish, or preparation of the
manuscript.

**Competing interests:** The authors have declared
that no competing interests exist.

childbirth) [2]. On the other hand, many women experience positive psychological outcomes after the event of childbirth, with 44.0% to 50.2% of women reporting moderate degrees of postpartum psychological growth [3,4]. Yet little is known about whether or how the experience of childbirth affects the social psychology of mothers. Previous research into the effects of sharing painful experiences with others suggests that it can lead to exceptionally strong bonds, effectively fusing together one's personal and group identities. Here we consider whether painful childbirth is associated with similar bonding.

## Childbirth as a catalyst for identity fusion

The process of 'identity fusion'—a visceral sense of 'oneness' with the group—can occur when perceived sharing of emotionally intense experiences with others creates a porous boundary between personal and group identities [5–7]. Could perceptions of shared experience among first-time mothers serve to fuse them to other mothers in this way? Previous studies of identity fusion, conducted in a wide range of special populations ranging from victims of terrorist attacks [e.g., 8] and survivors of natural disasters [9] to fans of football clubs [e.g., 10] and initiates into martial arts clubs [e.g., 11]–have shown that sharing of intense experiences can produce powerful social bonds associated with personally enhancing outcomes. Here we investigate whether analogous processes are at work in first-time mothers who have shared similar childbirth experiences with other mothers, whether the delivery of the child is traumatic or relatively stress-free, by measuring first-time mothers' perceptions of the childbirth experience, childbirth-related thought, and postpartum psychological health. Do shared experiences of childbirth bond women to one another in much the same way as 'bands of brothers' are forged on the battlefield [12]?

While frontline fighters serve shoulder to shoulder with other members of their battalion, childbirth is typically an individual event that is not physically shared with other mothers who are simultaneously experiencing labor, so mothers would need to rely on "indirect evidence of shared experience" [6, p. 11] to experience fusion. Whitehouse and colleagues [6,7,13] have argued that the underlying mechanism of identity fusion via shared experience is the formation of the autobiographical self via episodic memories thought to be shared with others. An episodic memory becomes a central component of one's autobiographical self; perceiving this core aspect in another individual then elicits kin detection, ultimately leading to "psychological kinship" [14]. Previous studies show that shared negative experiences, and some positive experiences, are associated with stronger levels of identity fusion among group members [e.g., 10,11]. For example, a greater proportion of frontline battalion fighters than non-fighters in the 2011 revolutionary war in Libya were more fused with their own battalion than family, suggesting that intense fighting experiences, when perceived as shared, cultivate identity fusion among battalion members [12].

Thought processes are also posited to play a key role in the relationship between shared negative experiences and identity fusion [8], but little is known about their exact nature (e.g., reflection or rumination), or whether identity fusion moderates the link between thought processes and psychological health (e.g., depression, PTSD, PTG).

## Factors predicting PTSD and PTG

A meta-analysis of 50 studies identified vulnerability factors before childbirth (e.g., antenatal depression, a fear of childbirth), risk factors during childbirth (e.g., lack of agency and negative emotions during the birth experience, an assisted vaginal or caesarean childbirth), and postpartum maintaining factors (postpartum depression, poor coping and stress) that place women at special risk of experiencing postpartum traumatic stress responses (PTSD) [15].

Factors that strengthen resiliency and lessen risk in pregnancy, during childbirth, and postpartum can take on a protective role, whereby women are empowered to acclimate and prosper after living through such a challenging experience [16].

Greater posttraumatic growth (PTG) with respect to childbirth is associated with a range of factors. These include greater use of approach-based coping (asking for guidance and support, taking problem-solving action) and avoidance-based coping (pursuing alternative rewards; i.e., engaging in alternative activities and fostering new sources of fulfillment), greater perceived social support, higher challenge appraisal (e.g., extent to which a woman believes that motherhood will help her know herself better), lower threat appraisal (e.g., extent to which a woman believes that motherhood will cause economic strain), and younger age. Somewhat counterintuitively, other predictors of PTG are higher PTSD symptoms during pregnancy, greater general distress after childbirth, an assisted vaginal or caesarean childbirth, and a preterm childbirth [e.g., 3,4,17–20].

Outside the childbirth experience, previous research spanning a range of potentially traumatic events has demonstrated differing relationships between event perception, coping styles, and negative and positive psychological outcomes. Akin to a "double-edged sword" that sanctions simultaneous debilitation and growth, the more central a traumatic event was construed to be to one's identity and life story, the greater the likelihood of PTSD symptoms (i.e., intrusive and avoidant thoughts), depression, and PTG. Strikingly, event centrality was the strongest unique predictor of PTSD and PTG symptoms, even after controlling for other factors such as depression, cognitive processing of the traumatic event, coping styles, and event severity [21]. Event centrality coupled with avoidant coping, a negative outlook on the event, and deeply emotional reactions to the event uniquely predicted PTSD symptoms, whereas event centrality coupled with problem-focused coping and a positive outlook on the event uniquely predicted PTG [22]. Notably, only social support and emotion-focused coping were unique predictors of PTG among women, where the more a woman perceived social encouragement and thought about minimizing or controlling her emotional suffering, the greater PTG she experienced [23].

Previous research also demonstrates that different forms of repetitive thought, whether they are voluntary or involuntary, trait-like [24–28] or event-related [see 29,30 for details], predict PTSD and PTG in noteworthy ways. In one study, when comparing trait-like repetitive thought (reflecting versus brooding) with event-related repetitive thought (deliberate versus intrusive), deliberate repetitive thought and event centrality were unique predictors of PTG, where the more an individual voluntarily tried to understand the meaning of the event and the more they viewed the event as important to their identity, the greater PTG they experienced. Brooding and event centrality were unique predictors of depressive symptoms, where the more an individual passively engaged in thinking about what is negative in their life and the more they perceived the event as important to their identity, the greater depressive symptoms they experienced. Finally, brooding, event centrality, and reflecting were unique predictors of PTSD symptoms, where the more an individual reflexively engaged in contemplating about what is bad in their life, the more they viewed the event as important to their identity, and—counterintuitively—the more they purposefully engaged in finding a solution to a problem, the greater PTSD symptoms they experienced [31].

In another study, when examining trait-like tendencies (reflecting versus ruminating), event-related repetitive thought (deliberate versus intrusive), and the perceived impact of the event, deliberate repetitive thought, ruminating, and core beliefs were unique predictors of PTG. The more an individual consciously thought about the significance of the event, the less they engaged in neurotic self-focus that had been provoked by perceived personal losses in the

past, and the more they believed that the event called into question their core beliefs about the world, the greater PTG they experienced. Intrusive repetitive thought was a unique predictor of distress symptoms, where the more an individual uncontrollably thought about the event, the greater distress symptoms they experienced. Notably, intrusive repetitive thought was the strongest unique predictor of deliberate repetitive thought, where the more an individual involuntarily thought about the event, the greater problem-solving and understanding of the event they experienced. It is possible that intrusive repetitive thoughts propel an individual to then deliberately and constructively find meaning after going through a traumatic experience [32]. Previous research suggests that thought processes about traumatic experiences can help to make them both personally transformative and group defining [33], laying the foundations for identity fusion [8].

## The present study

Our central aim was to investigate whether the experience of childbirth was related to identity fusion among first-time mothers. A secondary aim was to test potential event-related mechanisms of identity fusion, namely perceived sharedness, emotional intensity, centrality, and childbirth-related thoughts. A third aim was to understand whether first-time mothers who experience greater identity fusion via the event of childbirth enjoy psychological health benefits, such as a reduced risk of PTSD and postpartum depression, or a boost in PTG. Participants included first-time mothers aged 18 years and older who were either pregnant or had a child who was 6 months of age or younger. In line with the shared experience pathway to identity fusion [6,7], we examined first-time mothers within the first 6 months of their child's birth for two reasons: 1) to magnify the significance and novelty of the experience, and 2) to constrain the time period. Identity fusion measures using three different target groups ("all mothers", "mothers who have had a very difficult birth", and "mothers who have had an easy birth") were administered to all participants along with measures of perceived sharedness, anticipated/experienced pain, ruminative and reflective thought about the birth, the centrality of the childbirth event, and psychological health (i.e., depression, posttraumatic stress, and posttraumatic growth symptoms) adapted for each mother group.

Our hypotheses were as follows: 1) We predicted that fusion levels would be higher in the postpartum than antenatal group, because we hypothesized that first-time mothers needed to experience the event of childbirth to feel more strongly fused with other mothers; 2) Among postpartum mothers, we predicted that fusion with mothers who experienced a very difficult birth would be stronger for those who perceived the childbirth event as more shared, painful, and central; conversely, we predicted that fusion with mothers who experienced an easy birth would be stronger for those who perceived the childbirth event as less shared, painful, and central; 3) Among postpartum mothers, we predicted that greater childbirth dysphoria would be associated with more ruminative and reflective thought, which in turn would be associated with greater identity fusion with mothers who have had a very difficult birth; conversely, we predicted that less childbirth dysphoria would be associated with less ruminative and reflective thought, which in turn would be associated with greater identity fusion with mothers who have had an easy birth; 4) Among postpartum mothers, we predicted that identity fusion might inoculate against negative psychological health (depression and PTSD) and might promote PTG. Specifically, we predicted that fusion would moderate the relationship between event centrality, ruminative and reflective thought, and psychological health.

## Method

The study was conducted in accordance with and approved by the School of Anthropology and Museum Ethnography Departmental Research Ethics Committee (SAME DREC) at the University of Oxford.

### Participants

A total of 164 first-time mothers who were at least 18 years of age and living in the United States were recruited from November 2016 to February 2017 either via advertisements (see S1 Appendix) that were posted on various parenting and childbirth-related websites in compliance with their terms and conditions (e.g., www.bundoo.com, www.reddit.com; $n = 64$) or Qualtrics Panel Services ($n = 100$). First-time mothers were either antenatal ($n = 89$; between 4 to 40 weeks pregnant, $M = 26.47$ weeks, $SD = 8.82$; mothers 18 to 39 years), or postpartum ($n = 75$; with a baby between 1 to 28 weeks old, $M = 15.16$ weeks, $SD = 6.97$; mothers 18 to 45 years). We recruited mothers via parenting and childbirth-related websites (antenatal mothers; $n = 39$; postpartum mothers; $n = 25$) and via Qualtrics Panel Services (antenatal mothers; $n = 50$; postpartum mothers; $n = 50$).

### Procedure

Two online questionnaires were constructed using Qualtrics Online Survey Software; one questionnaire was designed for antenatal mothers and another for postpartum mothers. In an effort to make the two online questionnaires equivalent, both antenatal and postpartum mothers were provided with trauma and depression measures. All questionnaire responses were anonymous. Mothers who completed the online questionnaires via parenting and childbirth-related websites were emailed a $10 Amazon e-gift card.

### Antenatal questionnaire

The antenatal questionnaire took participants from 5 to 255 minutes to complete ($M = 16.83$ minutes, $SD = 29.57$). Only participants who indicated past trauma exposure were asked to provide details about the trauma; consequently, they took longer to complete the questionnaire. In order to ensure that only pregnant first-time mothers completed the antenatal questionnaire, participants were first provided with a screening question, "What describes your life stage best?", whereby antenatal mothers had to choose the option "Currently pregnant with first child". Refer to S2 Appendix for the screening question and answer choices. In order to make sure that participants were carefully reading the questionnaire, they were provided with an attention check question halfway through the questionnaire ("What color is the sky? Make sure to select orange for this answer so that we know you are reading the instructions of this survey"). Only respondents who answered "orange" were able to complete the questionnaire. After giving informed consent, participants provided demographic information that included their ethnicity, education level, marital status, and how far along they were in their pregnancy. The antenatal questionnaire items were presented in the order outlined below.

**Identity fusion.** Identity fusion was measured using the pictorial measure of fusion [34] with three different target groups, namely "all mothers", "mothers who have had a very difficult birth", and "mothers who have had an easy birth". The pictorial measure is comprised of five pictures, with each picture designated a letter (A, B, C, D, and E) and showing different degrees of overlap (0%, 25%, 50%, 75%, and 100%) between a smaller "self" circle and a larger "group" circle. Participants were asked to select one of the five pictures that best represented their relationship with the specified target group. Identity fusion was measured as a continuous

variable, whereby 0% overlap between the two circles was assigned a score of 1 (not fused) and 100% overlap between the two circles was assigned a score of 5 (strongly fused).

**Expected pain and difficulty of childbirth.**   Antenatal participants were asked to rate expected pain and difficulty of childbirth using two items developed for this study: "How painful do you expect your childbirth to be?" was measured on a 10-point scale (1 = not painful at all, 10 = very painful), and "How difficult do you expect your childbirth to be?" was measured on a 10-point scale (1 = not difficult at all, 10 = very difficult).

**Past trauma exposure.**   PTSD symptom frequency and severity in the last month was measured using the 24-item Posttraumatic Diagnostic Scale—Self-Report Version for DSM-5 (PDS-5) [35]. Participants were first provided with two trauma screening questions from the PDS-5, whereby only those who indicated past exposure to different traumatic experiences (e.g., physical assault, child abuse, natural disaster) and identified a single traumatic experience that was on their mind and currently bothered them were provided with twenty symptom items. The symptom items of the PDS-5 assess the frequency and severity of PTSD symptoms with respect to the single identified traumatic experience, and are based on the four DSM-5 symptom clusters, namely re-experiencing symptoms (items 1–5; e.g., "Unwanted upsetting memories about the trauma"), avoidance (items 6–7; e.g., "Trying to avoid thoughts or feelings related to the trauma"), changes in cognition and mood (items 8–14; e.g., "Not being able to remember important parts of the trauma"), and increased arousal and reactivity (items 15–20; e.g., "Acting more irritable or aggressive with others"). The symptom items were rated on a 5-point scale of frequency and severity (0 = not at all, 4 = 6 or more times a week/severe). Internal reliability was good, Cronbach's $\alpha$ = .96. Participants were also provided with four additional questions from the PDS-5 that relate to distress and interference caused by symptoms as well as symptom onset and duration. These four questions were not included in our analyses as PTSD symptom frequency and severity is measured by totalling the twenty PDS-5 symptom ratings [35].

**Antenatal depression.**   Antenatal depression symptom severity was measured using an adapted version of the 10-item Edinburgh Postnatal Depression Scale (EPDS) [36]. Previous studies have used the EPDS in order to screen for depression during a woman's pregnancy [37; cf. 38]. Instead of the phrase "As you have recently had a baby" in the instructions, our antenatal version substituted the phrase "As you are in the first, second, or third trimester of your pregnancy". Participants were then provided with the 10-item EPDS using a 4-point scale (0, 1, 2, and 3 according to increasing severity of symptom; e.g., 0 = no, not at all, 3 = yes, quite a lot). Internal reliability was acceptable, Cronbach's $\alpha$ = .87. Examples of items include: "I have been able to laugh and see the funny side of things" and "I have felt scared or panicky for no very good reason". Refer to S3 Appendix for the full adapted version of the 10-item EPDS.

## Postpartum questionnaire

The postpartum questionnaire took participants from 14 to 324 minutes to complete ($M$ = 38.40 minutes, $SD$ = 37.50). In order to ensure that only first-time mothers with babies 6 months of age or younger completed the postpartum questionnaire, participants were first provided with two screening questions: "What describes your life stage best?" and "What age group does your firstborn child fall under?", whereby postpartum mothers had to choose the options "Not currently pregnant, but have given birth to one child" and "0–6 months", respectively. Refer to S2 and S4 Appendices for the screening questions and answer choices. The attention check question and demographics were identical to the antenatal questionnaire, with the addition of a second attention check question as well as their child's age. The postpartum questionnaire items were presented in the order outlined below.

**Identity fusion.** Identity fusion was measured in the same format as in the antenatal questionnaire described above.

**Childbirth narrative.** Participants were asked to provide a childbirth narrative: "Describe your childbirth experience in as much detail as you can". This item was considered to be outside of the scope of the current paper and thus was not included in our analyses.

**Traumatic childbirth experience.** PTSD symptom frequency and severity in the last month with respect to the participant's childbirth experience was measured using an adapted version of the 24-item Posttraumatic Diagnostic Scale—Self-Report Version for DSM-5 (PDS-5) [35]. The instructions as well as the twenty symptom items of the PDS-5 include phrases like "the trauma" or "the traumatic event"; our version substituted the phrase "your childbirth experience". The symptom items were rated on a 5-point scale of frequency and severity (0 = not at all, 4 = 6 or more times a week/severe). Internal reliability was good, Cronbach's $\alpha$ = .92. Examples of symptom items include: "Unwanted upsetting memories about your childbirth experience", "Trying to avoid thoughts or feelings related to your childbirth experience", "Not being able to remember important parts of your childbirth experience", and "Blaming yourself or others (besides those involved in your childbirth experience) for what happened". The four additional questions from the PDS-5 that relate to distress and interference caused by symptoms as well as symptom onset and duration were not included in our analyses [35]. Refer to S5 Appendix for the full adapted version of the 24-item PDS-5.

**Postpartum depression.** Postpartum depression symptom severity was measured using the 10-item Edinburgh Postnatal Depression Scale (EPDS) [36]. The EPDS items were rated on a 4-point scale (0, 1, 2, and 3 according to increasing severity of symptom; e.g., 0 = no, not at all, 3 = yes, quite a lot). Internal reliability was good, Cronbach's $\alpha$ = .90.

**Pain of childbirth.** Participants were asked to rate pain of childbirth using four items developed for this study: "Overall, how intense was the pain of childbirth?" was measured on a 10-point scale (1 = not intense at all, 10 = very intense), "Overall, how unpleasant was the pain of childbirth?" was measured on a 10-point scale (1 = not unpleasant at all, 10 = very unpleasant), "At the peak of the pain, how intense was the pain of childbirth?" was measured on a 10-point scale (1 = not intense at all, 10 = very intense), and "At the peak of the pain, how unpleasant was the pain of childbirth?" was measured on a 10-point scale (1 = not unpleasant at all, 10 = very unpleasant). These items displayed moderate to very strong correlations with one another, ranging from $r$ = .68, $p$ < .001 to $r$ = .88, $p$ < .001. Internal reliability was good, Cronbach's $\alpha$ = .94. Consequently, the four pain items were combined to provide an average pain score.

**Posttraumatic growth following childbirth.** PTG after childbirth was measured using an adapted version of the 21-item Posttraumatic Growth Inventory (PTGI) [39]. Instead of the word "crisis" in the instructions and the 6-point Likert response format (e.g., "Indicate for each of the statements below the degree to which this change occurred in your life as a result of your crisis"), our version substituted the phrase "experience of childbirth" in the instructions as well as the word "experience" in the response choices. The PTGI items were rated on a 6-point scale (0 = I did not experience this change as a result of my experience, 5 = I experienced this change to a very great degree as a result of my experience). Internal reliability was good, Cronbach's $\alpha$ = .93. Examples of items include: "An appreciation for the value of my own life" and "Knowing that I can count on people in times of trouble". Refer to S6 Appendix for the full adapted version of the 21-item PTGI.

**Perceived sharedness of childbirth.** Perceived sharedness of childbirth was measured using two items developed for this study: "How painful do you think childbirth has been for most other mothers?" was measured on a 10-point scale (1 = not painful at all, 10 = very painful), and "Do you think your childbirth has been more or less painful than what you think to

be a typical childbirth experience?" was measured on a 10-point scale (1 = a lot less painful, 10 = a lot more painful).

**Ruminative and reflective thought.** Ruminative and reflective thought was measured using an adapted version of the 20-item Event Related Rumination Inventory (ERRI) [32]. The ERRI consists of two 10-item subscales—one subscale measures intrusive repetitive thinking about a highly distressing event, and the other subscale measures deliberate repetitive thinking about a highly distressing event. The ERRI is analogous to other measures that distinguish between different forms of repetitive thought (e.g., reflection versus brooding; Ruminative Responses Scale (RRS) [27]; reflection versus rumination; Rumination-Reflection Questionnaire (RRQ) [28]); nevertheless, the ERRI is uniquely tailored to examine specific event-related repetitive thought with respect to a highly distressing life experience. In an effort to clarify the distinction, the current study utilizes the intrusive subscale to measure what is typically thought of as rumination and the deliberate subscale to measure what is typically thought of as reflection. The instructions for both the intrusive and deliberate items of the ERRI begin with the phrase "After an experience like the one you reported", and end with the phrase "during the weeks immediately after the event"; in our version, the instructions for both the intrusive and deliberate items began with the phrase "After an experience like the one you reported (your experience of childbirth)", and ended with the phrase "during the weeks immediately after your experience of childbirth". The intrusive and deliberate items of the ERRI were rated on a 4-point scale (0 = not at all, 3 = often). Internal reliability was good for both the intrusive and deliberate subscales, Cronbach's $\alpha$ = .95 and .91, respectively. Examples of items include: "I thought about the event when I did not mean to" for the intrusive subscale; "I thought about whether I could find meaning from my experience" for the deliberate subscale. Refer to S7 Appendix for the full adapted version of the 20-item ERRI.

**Extent of thought about childbirth.** Extent of thought about childbirth was measured using two items developed for this study: "How much do you think about the meaning of your childbirth?" and "How much do you think about how your childbirth could have turned out differently (e.g., how it could have been easier, or how it could have been worse)?". Both items were measured on a 6-point scale (0 = I have only thought about it a bit, 5 = it is always on my mind).

**Centrality of child's birth.** Centrality of child's birth was measured using an adapted version of the short 7-item Centrality of Event Scale (CES) [40]. Instead of the phrase "the most stressful or traumatic event in your life" ("Please think back upon the most stressful or traumatic event in your life") in the instructions, or the phrase "this event" in the individual items, our version substituted the phrase "your child's birth" in the instructions as well as "my child's birth" in the individual items. The CES items were rated on a 5-point scale (1 = totally disagree, 5 = totally agree). Internal reliability was acceptable, Cronbach's $\alpha$ = .89. Examples of items include: "I feel that my child's birth has become part of my identity" and "My child's birth has colored the way I think and feel about other experiences". Refer to S8 Appendix for the full adapted version of the short 7-item CES.

## Results

No data were missing as participants were required to complete all of the questionnaire items. For the antenatal group, 46 out of 89 first-time mothers (51.7%) indicated past exposure to traumatic experiences and completed the PDS-5. Table 1 contains descriptive information for the potential identity fusion mechanism and psychological health variables by mother group.

**Table 1. Means (and SDs) for potential identity fusion mechanisms and psychological health as a function of mother group.**

| Measure | Antenatal Mothers (n = 89) | Postpartum Mothers (n = 75) |
|---|---|---|
| **Potential Identity Fusion Mechanisms** | | |
| Perceived sharedness of childbirth | | |
| 1. "How painful do you think childbirth has been for most other mothers?" | — | 7.92 (2.03) |
| 2. "Do you think your childbirth has been more or less painful than what you think to be a typical childbirth experience?" | — | 4.73 (2.77) |
| Pain of childbirth (an average of the four pain items) | — | 7.05 (2.77) |
| Expected pain and difficulty of childbirth | | |
| 1. "How painful do you expect your childbirth to be?" | 8.00 (1.86) | — |
| 2. "How difficult do you expect your childbirth to be?" | 6.74 (2.18) | — |
| CES | — | 26.44 (7.09) |
| ERRI-I | — | 9.20 (8.33) |
| ERRI-D | — | 11.24 (8.01) |
| Extent of thought about childbirth | | |
| 1. "How much do you think about the meaning of your childbirth?" | — | 2.08 (1.58) |
| 2. "How much do you think about how your childbirth could have turned out differently (e.g., how it could have been easier, or how it could have been worse)?" | — | 2.20 (1.69) |
| **Psychological Health** | | |
| EPDS | 9.10 (5.19) | 8.04 (5.73) |
| PDS-5[†] | 17.74 (16.64) [‡] | 11.17 (11.02) |
| PTGI | — | 57.44 (22.48) |

CES, Centrality of Event Scale; ERRI-I, Event Related Rumination Inventory—Intrusive (ruminative thought); ERRI-D, Event Related Rumination Inventory—Deliberate (reflective thought); EPDS, Edinburgh Postnatal Depression Scale; PDS-5, Posttraumatic Diagnostic Scale—Self-Report Version for DSM-5; PTGI, Posttraumatic Growth Inventory.

[†] With respect to a past traumatic experience for antenatal mothers and with respect to the childbirth experience for postpartum mothers.

[‡] N = 46 for antenatal mothers.

## Preliminary analyses

**Sample characteristics.** Table 2 contains analyses of differences between antenatal mothers and postpartum mothers for demographic characteristics. There were no significant differences between the two groups for age, level of education, ethnicity, or marital status. Because antenatal mothers differed in their weeks of pregnancy, and postpartum mothers differed in the age of their baby, we also conducted correlations between these two variables, identity fusion, and potential identity fusion mechanisms. There were no significant correlations between the three fusion measures and weeks of pregnancy (antenatal; $r$s ranged from -.002 to .11, n.s.), or between the three fusion measures and age of baby (postpartum; $r$s ranged from -.09 to .11, n.s.). For the antenatal group, there were no significant correlations between weeks of pregnancy and expected pain and difficulty ($r$s from -.002 to .09, n.s.). For the postpartum group, there were no significant correlations between age of baby and perceived sharedness, pain, and the thinking measures ($r$s ranged from -.03 to .17, n.s.).

**Table 2. Demographic characteristics of antenatal and postpartum mothers.**

| Demographic Characteristics | Antenatal Mothers (n = 89) Means (and SDs) or Frequency | Postpartum Mothers (n = 75) Means (and SDs) or Frequency | Statistics |
|---|---|---|---|
| Age in years | 28.11 (5.19) | 27.96 (5.67) | $t(162) = .18, p = .86$ |
| Highest level of education completed | | | $\chi^2(1) = 2.31, p = .13$ |
| a. Less than a bachelor's degree | 33.7% | 45.3% | |
| b. Bachelor's degree and higher | 66.3% | 54.7% | |
| Ethnicity of the participant | | | $\chi^2(1) = .01, p = .93$ |
| a. White—non Hispanic | 75.3% | 74.7% | |
| b. Not White—non Hispanic | 24.7% | 25.3% | |
| Marital status of the participant | | | $L\chi^2(4) = 3.41, p = .49$ |
| a. Single | 10.1% | 6.7% | |
| b. Living with their partner | 16.9% | 17.3% | |
| c. Married | 71.9% | 74.7% | |
| d. Divorced | 1.1% | 0.0% | |
| e. Widowed | 0.0% | 1.3% | |

**Intercorrelations among potential identity fusion mechanisms.** For the antenatal group, expected pain and difficulty of childbirth were moderately correlated with one another, $r = .55, p < .001$. For the postpartum group, the second perceived sharedness question and pain were both positively correlated with ruminative and/or reflective thought. Intercorrelations among the potential identity fusion mechanism variables for postpartum mothers are listed in Table 3.

**Intercorrelations among psychological health measures.** As demonstrated in previous research [e.g., 35,41], in both groups there were significant positive links between PTSD symptoms and depression symptoms ($r$s from .55 to .78, $p$s < .001). That is, antenatal mothers who reported higher levels of PTSD symptom frequency and severity with reference to a past

**Table 3. Correlations among potential identity fusion mechanisms for postpartum mothers.**

| | 1 | 2 | 3 | 4 | 5 | 6 | 7 | 8 |
|---|---|---|---|---|---|---|---|---|
| 1. Shared1 | — | -.14 | .28* | .06 | -.05 | .15 | .11 | .01 |
| 2. Shared2 | | — | .65*** | .09 | .27* | .33** | .19 | .25* |
| 3. Pain | | | — | .06 | .26* | .18 | .07 | .21 |
| 4. CES | | | | — | .27* | .46*** | .48*** | .29* |
| 5. ERRI-I | | | | | — | .54*** | .31** | .55*** |
| 6. ERRI-D | | | | | | — | .56*** | .50*** |
| 7. Thought1 | | | | | | | — | .38*** |
| 8. Thought2 | | | | | | | | — |

Shared1, perceived sharedness of childbirth (item 1); Shared2, perceived sharedness of childbirth (item 2); Pain, pain of childbirth (an average of the four pain items); CES, Centrality of Event Scale; ERRI-I, Event Related Rumination Inventory—Intrusive (ruminative thought); ERRI-D, Event Related Rumination Inventory—Deliberate (reflective thought); Thought1, extent of thought about childbirth (item 1); Thought2, extent of thought about childbirth (item 2).

*$p < .05$.

**$p < .01$.

***$p < .001$.

traumatic experience had higher levels of antenatal depression symptom severity, and postpartum mothers who reported higher levels of PTSD symptom frequency and severity with respect to their childbirth experience had higher levels of postpartum depression symptom severity. For the postpartum group, in line with previous studies [e.g., 3,42; cf. 4,21,31,43], there was no link between PTSD symptom frequency and severity and PTG ($r$ = .15, n.s.).

### Main analyses

**Identity fusion among antenatal and postpartum mothers.** Our first hypothesis was that first-time mothers who had already given birth (postpartum mothers) would feel more strongly fused with other mothers compared to first-time mothers who had not yet given birth (antenatal mothers). We conducted separate one-way analyses of variance (ANOVAs) to test the effect of mother group on the three fusion targets: all mothers, mothers who have had a very difficult birth, and mothers who have had an easy birth. When homogeneity of variance was violated (for the difficult birth fusion target), the Welch statistic was used. There was a main effect of mother group on identity fusion with all mothers ($F(1, 162)$ = 10.60, $p$ = .001, $\eta_p^2$ = .06); postpartum mothers ($M$ = 3.84, SD = 1.20) fused significantly higher with all mothers than did antenatal mothers ($M$ = 3.18, SD = 1.37). There was also a main effect of mother group on identity fusion with mothers who have had a very difficult birth ($F(1, 140.40)$ = 7.07, $p$ = .009, $\eta_p^2$ = .04, $\omega^2$ = .04); postpartum mothers ($M$ = 2.29, SD = 1.42) fused significantly higher with mothers who have had a very difficult birth than did antenatal mothers ($M$ = 1.75, SD = 1.13). Finally, there was a main effect of mother group on identity fusion with mothers who have had an easy birth ($F(1, 162)$ = 10.23, $p$ = .002, $\eta_p^2$ = .06); postpartum mothers ($M$ = 3.05, SD = 1.55) fused significantly higher with mothers who have had an easy birth than did antenatal mothers ($M$ = 2.30, SD = 1.45). Fig 1 shows the scores for the three fusion targets reported by each mother group.

**Links between potential mechanisms and identity fusion among postpartum mothers.** Our second hypothesis was that, among postpartum mothers, identity fusion would be correlated with perceived sharedness, emotional intensity, centrality, and childbirth-related thoughts. Specifically, we hypothesized that perceived sharedness, pain levels, centrality of childbirth, and childbirth-related thought would positively correlate with fusion for mothers who have had a very difficult birth, and negatively correlate with fusion for mothers who have had an easy birth. Table 4 shows that higher identity fusion with mothers who have had a very difficult birth was indeed associated with higher levels of perceived sharedness and childbirth-related thoughts including ruminative and reflective thought, whereas identity fusion with mothers who have had an easy birth showed the inverse pattern for most of these same mechanisms. However, centrality of the childbirth event was not correlated with fusion for any of the three fusion targets; nor were any of the mechanisms correlated with fusion to all mothers.

To assess the unique contributions of these potential mechanisms to fusion, we conducted separate hierarchical linear regression analyses for the fusion targets of "mothers who have had a very difficult birth" and "mothers who have had an easy birth". Potentially confounding demographic variables were identified using correlation analyses and entered in the first step; in the second step of the model, perceived sharedness of childbirth (item 2), ruminative and reflective thought, and extent of thought about childbirth (item 2) were entered. In the final model (see Table 5), both item 2 of perceived sharedness of childbirth and ruminative thought uniquely predicted fusion, such that thinking that your childbirth was more painful than a typical childbirth experience and more ruminative thinking about the birth predicted greater identity fusion with mothers who have had a very difficult birth.

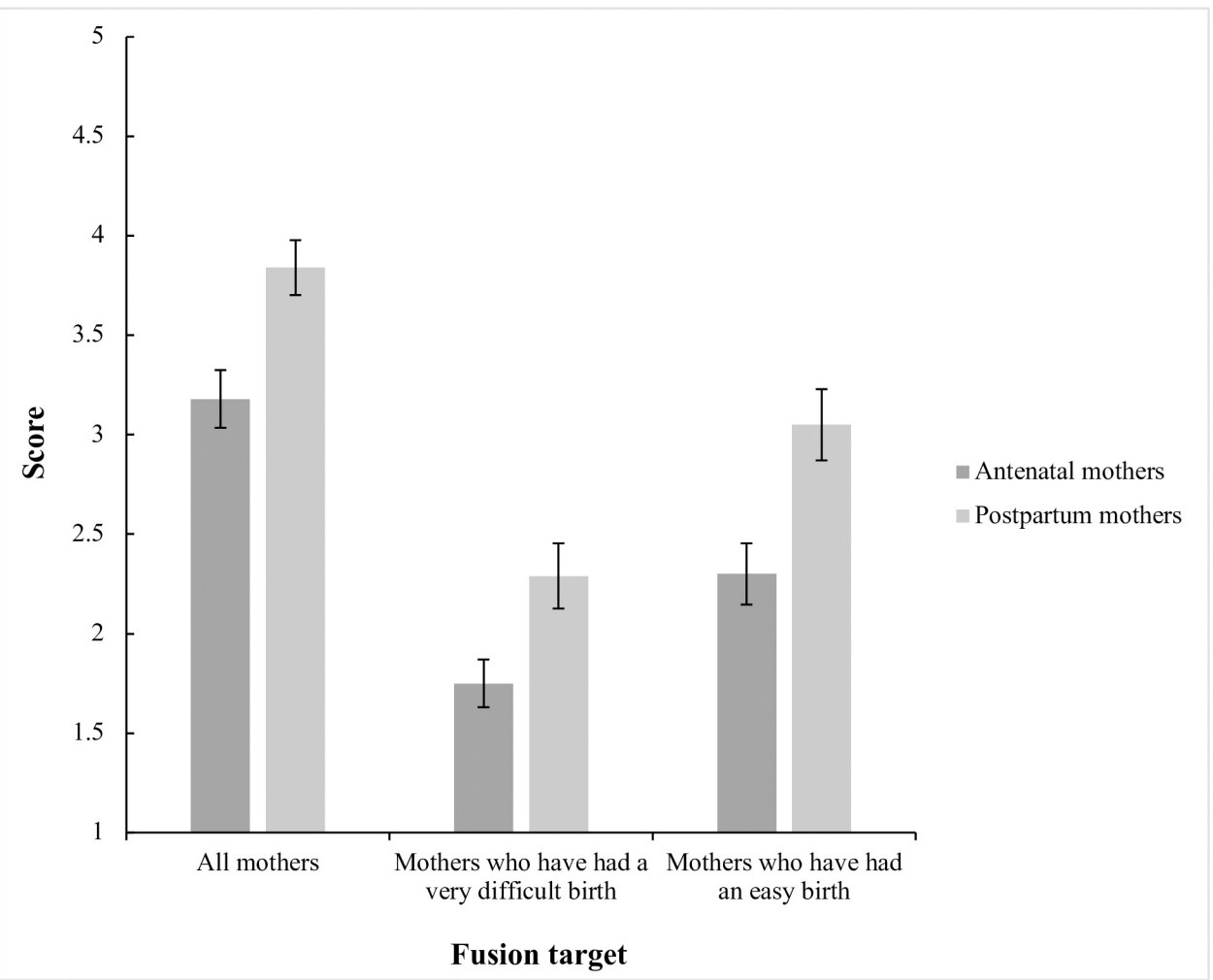

**Fig 1. Scores for the three fusion targets as a function of mother group.**

The same process was used for the hierarchical linear regression analysis with identity fusion with mothers who have had an easy birth. In the second step of the model, perceived sharedness of childbirth (item 2), pain of childbirth, ruminative thought, and extent of thought about childbirth (item 2) were entered. In the final model (see Table 6), both age and item 2 of perceived sharedness of childbirth uniquely predicted fusion, such that younger age and thinking that your childbirth was less painful than a typical childbirth experience predicted greater identity fusion with mothers who have had an easy birth.

**Repetitive thought about childbirth mediates the relationship between dysphoria and identity fusion among postpartum mothers.** Our third hypothesis was that, among postpartum mothers, greater childbirth dysphoria would be associated with more ruminative and reflective thought, which in turn would be associated with greater identity fusion with mothers who have had a very difficult birth. In turn, we expected that lower childbirth dysphoria would be associated with less ruminative and reflective thought, which in turn would be associated with greater identity fusion with mothers who have had an easy birth. We conducted separate mediation analyses using ordinary least squares path analysis in Hayes' [44] PROCESS macro

**Table 4. Pearson correlations between potential mechanisms and the three fusion targets.**

| Potential Identity Fusion Mechanisms | Fusion Target | | |
|---|---|---|---|
| | Difficult | Easy | All |
| Shared1 | -.22 | .01 | .11 |
| Shared2 | .48*** | -.54*** | .04 |
| Pain | .14 | -.27* | -.08 |
| CES | .09 | -.12 | -.05 |
| ERRI-I | .46*** | -.29* | -.15 |
| ERRI-D | .24* | -.07 | .03 |
| Thought1 | .10 | -.07 | .11 |
| Thought2 | .37*** | -.24* | -.02 |

Shared1, perceived sharedness of childbirth (item 1); Shared2, perceived sharedness of childbirth (item 2); Pain, pain of childbirth (an average of the four pain items); CES, Centrality of Event Scale; ERRI-I, Event Related Rumination Inventory—Intrusive (ruminative thought); ERRI-D, Event Related Rumination Inventory—Deliberate (reflective thought); Thought1, extent of thought about childbirth (item 1); Thought2, extent of thought about childbirth (item 2).

*$p < .05$.

**$p < .01$.

***$p < .001$.

(Model 4) for SPSS. Bias-corrected bootstrap analyses based on 10,000 bootstrap samples were run.

In the first model predicting fusion with mothers who have had a very difficult birth, when childbirth dysphoria was operationalized as item 2 of perceived sharedness, we found that thinking that one's childbirth was more painful than a typical childbirth experience was associated with more ruminative thought (denoted by the "a path" in Fig 2), $b = 0.80$, $t(73) = 2.37$, $p = .02$, and more ruminative thought was associated with greater identity fusion with mothers who have had a very difficult birth (denoted by the "b path" in Fig 2), $b = 0.06$, $t(72) = 3.63$, $p < .001$. There was also evidence that the extent to which thinking that one's childbirth was more or less painful than a typical childbirth experience was linked to identity fusion with

**Table 5. Final hierarchical multiple linear regression model on identity fusion with mothers who have had a very difficult birth.**

| Variable | | | | Difficult | | | | | |
|---|---|---|---|---|---|---|---|---|---|
| | B | SE | β | $R^2$ (Adj. $R^2$) | $R^2$ change | F | df | F change | df |
| | | | | .44 (.39) | .36 | 8.80*** | 6, 68 | 10.84*** | 4, 68 |
| Constant | -0.92 | 0.84 | — | | | | | | |
| Age in years | 0.05 | 0.03 | .21 | | | | | | |
| Highest level of education | 0.30 | 0.31 | .11 | | | | | | |
| Shared2 | 0.18 | 0.05 | .35*** | | | | | | |
| ERRI-I | 0.05 | 0.02 | .29* | | | | | | |
| ERRI-D | -0.00 | 0.02 | -.02 | | | | | | |
| Thought2 | 0.15 | 0.10 | .18 | | | | | | |

Shared2, perceived sharedness of childbirth (item 2); ERRI-I, Event Related Rumination Inventory—Intrusive (ruminative thought); ERRI-D, Event Related Rumination Inventory—Deliberate (reflective thought); Thought2, extent of thought about childbirth (item 2).

*$p < .05$.

**$p < .01$.

***$p < .001$.

**Table 6. Final hierarchical multiple linear regression model on identity fusion with mothers who have had an easy birth.**

| Variable | B | SE | β | $R^2$ (Adj. $R^2$) | $R^2$ change | F | df | F change | df |
|---|---|---|---|---|---|---|---|---|---|
| | | | | .50 (.45) | .32 | 11.13*** | 6, 68 | 10.88*** | 4, 68 |
| Constant | 7.18 | 0.88 | — | | | | | | |
| Age in years | -0.09 | 0.03 | -.34** | | | | | | |
| Highest level of education | -0.34 | 0.32 | -.11 | | | | | | |
| Shared2 | -0.31 | 0.07 | -.56*** | | | | | | |
| Pain | 0.08 | 0.07 | .15 | | | | | | |
| ERRI-I | -0.02 | 0.02 | -.13 | | | | | | |
| Thought2 | -0.10 | 0.10 | -.11 | | | | | | |

Shared2, perceived sharedness of childbirth (item 2); Pain, pain of childbirth (an average of the four pain items); ERRI-I, Event Related Rumination Inventory—Intrusive (ruminative thought); Thought2, extent of thought about childbirth (item 2).

*$p < .05$.

**$p < .01$.

***$p < .001$.

mothers who have had a very difficult birth when controlling for ruminative thought (denoted by the "c' path" in Fig 2), $b = 0.20$, $t(72) = 3.84$, $p < .001$.

We found a similar pattern when childbirth dysphoria was operationalized as pain levels: greater pain of childbirth was associated with more ruminative thought (denoted by the "a path" in Fig 3), $b = 0.78$, $t(73) = 2.31$, $p = .02$, and more ruminative thought was associated with greater identity fusion with mothers who have had a very difficult birth (denoted by the "b path" in Fig 3), $b = 0.08$, $t(72) = 4.20$, $p < .001$. There was no evidence that pain of childbirth was linked to identity fusion with mothers who have had a very difficult birth when controlling for ruminative thought (denoted by the "c' path" in Fig 3), $b = 0.01$, $t(72) = 0.20$, $p = .84$.

In the second model predicting fusion with mothers who have had an easy birth, we found that lower pain of childbirth was associated with less ruminative thought (denoted by the "a path" in Fig 4), $b = 0.78$, $t(73) = 2.31$, $p = .02$, and less ruminative thought was associated with greater identity fusion with mothers who have had an easy birth (denoted by the "b path" in Fig 4), $b = -0.04$, $t(72) = -2.07$, $p = .04$. There was no evidence that pain of childbirth was linked

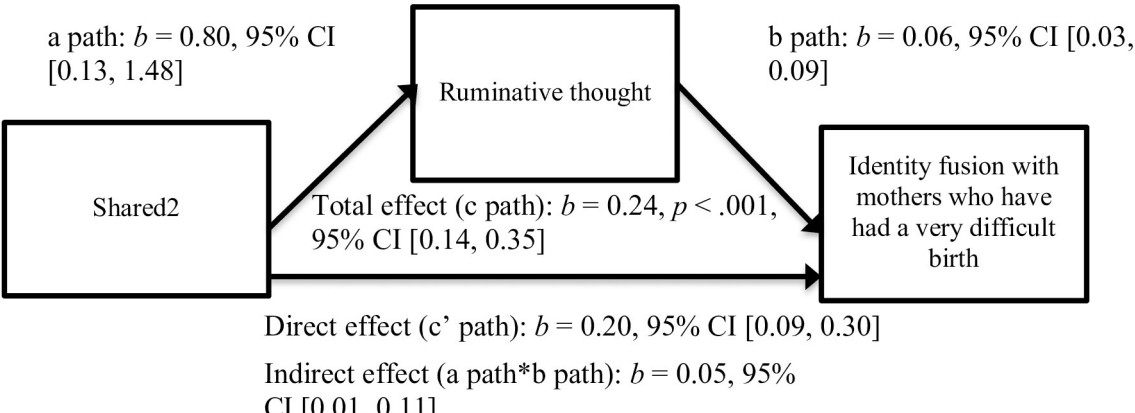

**Fig 2. Mediation model of fusion with mothers who had a difficult birth via sharedness and ruminative thought.**

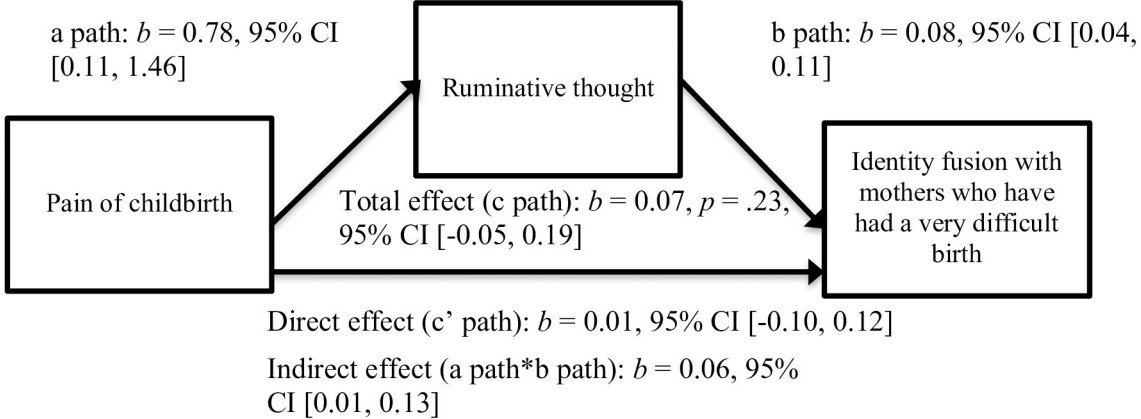

a path: *b* = 0.78, 95% CI [0.11, 1.46]

b path: *b* = 0.08, 95% CI [0.04, 0.11]

Ruminative thought

Pain of childbirth

Total effect (c path): *b* = 0.07, *p* = .23, 95% CI [-0.05, 0.19]

Identity fusion with mothers who have had a very difficult birth

Direct effect (c' path): *b* = 0.01, 95% CI [-0.10, 0.12]

Indirect effect (a path*b path): *b* = 0.06, 95% CI [0.01, 0.13]

**Fig 3. Mediation model of fusion with mothers who had a difficult birth via pain levels and ruminative thought.**

to identity fusion with mothers who have had an easy birth when controlling for ruminative thought (denoted by the "c' path" in Fig 4), *b* = -0.12, *t*(72) = -1.82, *p* = .07.

**Identity fusion as a moderator of psychological health after childbirth.** Our fourth hypothesis was that, among postpartum mothers, identity fusion would moderate the relationship between event centrality, repetitive thoughts (rumination or reflection), and psychological health measures of depression, PTSD, and PTG. Specifically, we predicted that higher levels of fusion might inoculate against negative psychological health (depression and PTSD) and promote positive psychological health (PTG). We conducted moderation analyses using Hayes' [44] PROCESS macro (Model 1) for SPSS.

None of the models predicting depression or PTSD were significant, but two models predicting PTG were significant. First, the interaction between ruminative thought and identity fusion with all mothers was significant, *b* = 0.47, *t*(71) = 2.07, *p* = .04, indicating that the relationship between ruminative thought and PTG was moderated by identity fusion with all mothers (see Fig 5). When identity fusion with all mothers was low or average, there was no relationship between ruminative thought and PTG, *b* = -0.04, *t*(71) = -0.09, *p* = .93, *b* = 0.52, *t*(71) = 1.63, *p* = .11, respectively. When identity fusion with all mothers was high, there was a significant relationship between ruminative thought and PTG, *b* = 1.07, *t*(71) = 2.58, *p* = .012.

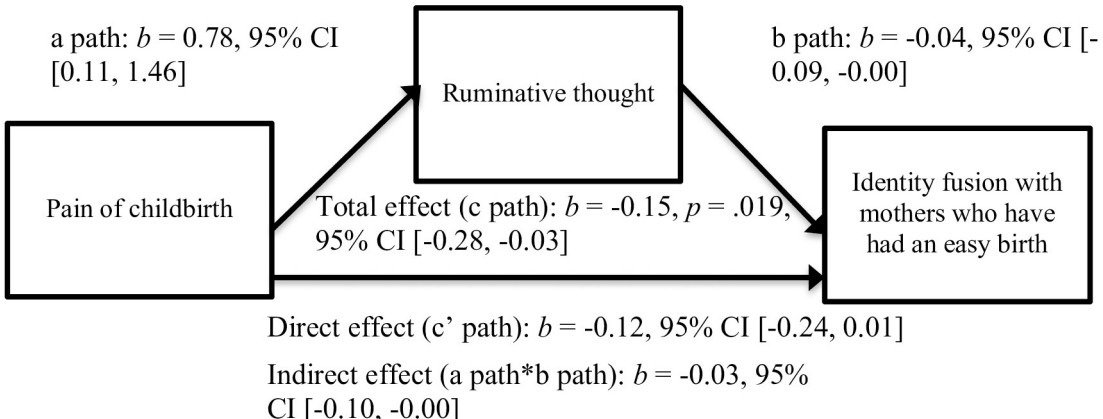

a path: *b* = 0.78, 95% CI [0.11, 1.46]

b path: *b* = -0.04, 95% CI [-0.09, -0.00]

Ruminative thought

Pain of childbirth

Total effect (c path): *b* = -0.15, *p* = .019, 95% CI [-0.28, -0.03]

Identity fusion with mothers who have had an easy birth

Direct effect (c' path): *b* = -0.12, 95% CI [-0.24, 0.01]

Indirect effect (a path*b path): *b* = -0.03, 95% CI [-0.10, -0.00]

**Fig 4. Mediation model of fusion with mothers who had an easy birth via pain levels and ruminative thought.**

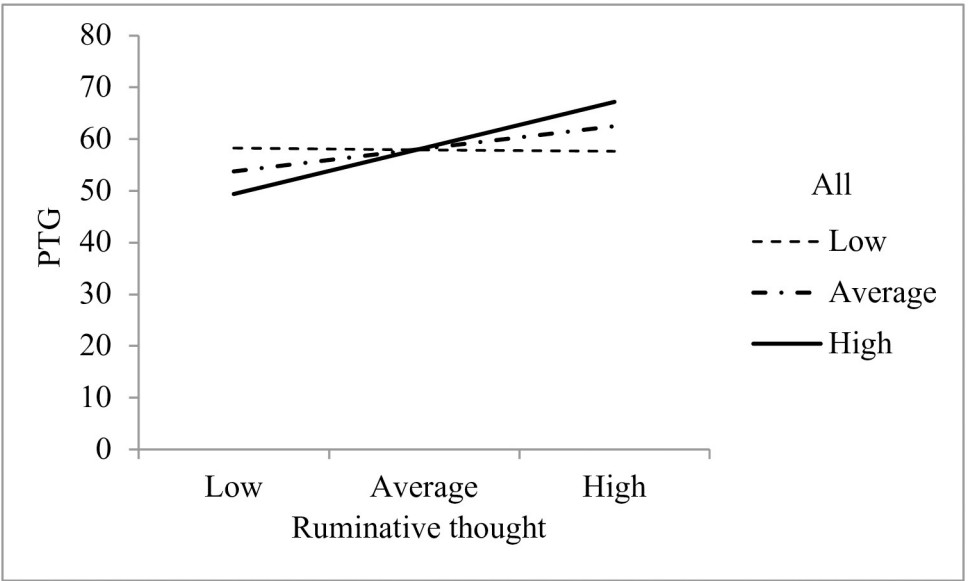

**Fig 5. Simple slopes of ruminative thought predicting PTG as a function of fusion with all mothers.**

That is, when postpartum mothers reported high identity fusion with all mothers, a 1-point increase in ruminative thought gave a 1.07-point increase in PTG.

The interaction between reflective thought and identity fusion with mothers who have had a very difficult birth was also significant, $b = 0.40$, $t(71) = 2.48$, $p = .016$, suggesting that the relationship between reflective thought and PTG was moderated by identity fusion with mothers who have had a very difficult birth (see Fig 6). At all levels of identity fusion, there was a significant relationship between reflective thought and PTG, $b = 1.03$, $t(71) = 2.70$, $p = .009$, $b = 1.55$, $t(71) = 5.07$, $p < .001$, and $b = 2.12$, $t(71) = 5.72$, $p < .001$, respectively. That is, when postpartum mothers reported low identity fusion with mothers who have had a very difficult birth, a 1-point increase in reflective thought gave a 1.03-point increase in PTG; when postpartum mothers reported average identity fusion with mothers who have had a very difficult birth, a 1-point increase in reflective thought gave a 1.55-point increase in PTG; when postpartum mothers reported high identity fusion with mothers who have had a very difficult birth, a 1-point increase in reflective thought gave a 2.12-point increase in PTG. Thus, the protective effect of fusion for PTG was greatest for mothers at the highest level of fusion with other mothers who had a difficult birth.

## Discussion

The current study investigated identity fusion in women who were either pregnant with their first child or had recently experienced childbirth for the first time. Our findings indicated that after giving birth, first-time mothers reported feeling more strongly fused with other mothers compared to first-time mothers who had not yet given birth. Among postpartum mothers, the main mechanisms of fusion were perceived sharedness, ruminative thought, and age. Rumination, but not reflection, played a mediating role between childbirth dysphoria and fusion levels. Lastly, higher levels of fusion helped mothers who ruminated and reflected more upon childbirth to experience greater PTG.

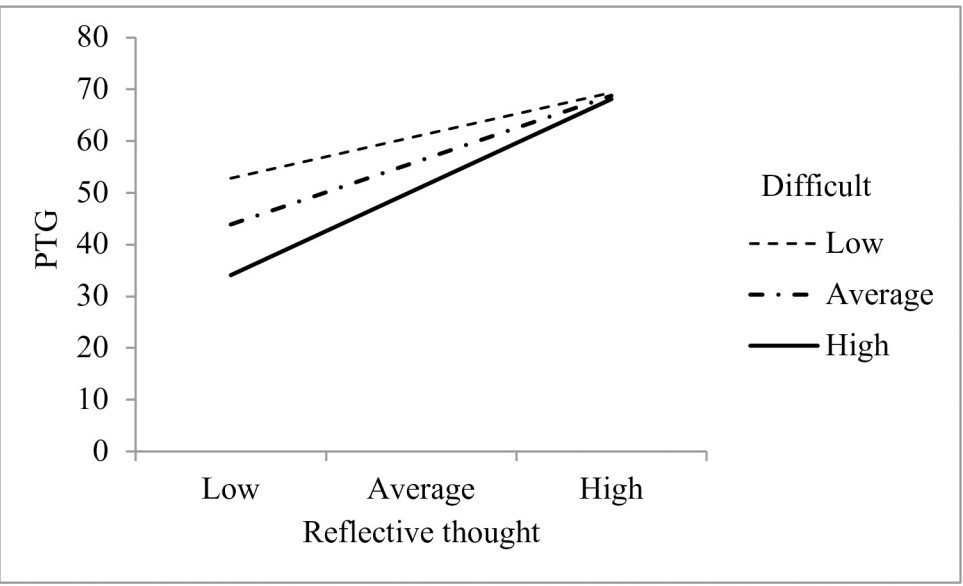

**Fig 6. Simple slopes of reflective thought predicting PTG as a function of fusion with mothers who had a difficult birth.**

## Identity fusion before and after childbirth

Our first hypothesis was supported: we found that identity fusion with other mothers was stronger in women who had experienced the event of childbirth for the first time. Whereas pregnant mothers drew upon their expectations and hopes for the upcoming novel event, post-partum mothers capitalized on their firsthand childbirth experience. Postpartum mothers have access to a detailed autobiographical memory that consists of the exact minutes, hours, or days in which their firstborn arrived into the world; they now belong to a symbolic sisterhood of women who have gone through similar childbirth experiences. These results are in line with previous research suggesting that identity fusion can result from highly emotional events that are perceived as shared with other group members [e.g., 5–8,10–13,45–48]. Nevertheless, it is important to consider that the magnitudes of the effects were small to medium, suggesting that perhaps there is a distinction to be made between the strength of extended fusion and local fusion. While indirect forms of sharedness can trigger extended fusion—in this case the indi-vidualistic experience of childbirth—local fusion may be cultivated by equally poignant events that are instead experienced side-by-side with other group members. Consequently, local fusion could be a more potent form and a stronger driver of prosocial outcomes [see also 5,6].

## Mechanisms of identity fusion

The current study offered three variations of a single fusion target: mothers. Our aim was to identify the effects of shared experience on first-time mothers' fusion to other mothers who had faced a momentous event akin to their own, but with varying degrees of intensity and overlap. Our second hypothesis was partially supported: fusion with mothers who endured a very difficult birth was stronger for postpartum mothers who exhibited higher levels of per-ceived sharedness and ruminative thought. This result suggests that postpartum mothers who feel that they have endured more pain and discomfort than the average mother, and who uncontrollably think about their birth event, perceive their childbirth as shared with other mothers who have also undergone a particularly stressful and demanding childbirth. Although

previous fusion studies have demonstrated that thought processes triggered by shared intense experiences can act as a precursor to fusion [see 7,8,45], we were able to identify their exact form.

We also found that fusion with mothers who underwent an easy birth was stronger for postpartum mothers who reported lower perceived sharedness. Postpartum mothers who feel that they have experienced less pain and discomfort than the average mother perceive their childbirth as shared with other mothers who have also gone through a straightforward and undemanding childbirth. Therefore, the absence of painful experiences, and not necessarily the presence of positive events [e.g., 10], is also associated with greater identity fusion.

We also discovered that fusion with mothers who experienced an easy birth was stronger for younger postpartum mothers. This finding suggests that younger postpartum mothers perceive their experience of childbirth as shared with other mothers who have gone through a trouble-free and uncomplicated childbirth. Previous research has shown that regular fetal screening tests during pregnancy and a history of infertility or pregnancy loss can contribute to older mothers (i.e., women in their mid 30s and early 40s) having a higher perception of pregnancy risk than younger mothers [e.g., 49,50]; this feeling sometimes translates to older mothers reporting more difficulties during and after childbirth than younger mothers [e.g., 51–53; cf. 54]. Older mothers are also more likely than younger mothers to indicate difficulties adapting to life after the arrival of their newborn, whereby they find their loss of independence and new lifestyle challenging [e.g., 55].

Overall, centrality of the childbirth event was not linked to fusion for any of our three fusion targets. This finding may be unique to the experience of childbirth, because past research has shown that the degree of self-shapingness is related to fusion in other kinds of contexts [see 10,13,45].

Our third hypothesis was partially supported: we demonstrated that postpartum mothers who endured greater childbirth dysphoria (by way of perceived sharedness or pain levels) engaged in more ruminative thought, and postpartum mothers who engaged in more ruminative thought fused more strongly with mothers who have had a very difficult birth. These results replicated and built on the assertion that shared dysphoria is associated with identity fusion via thought processes [8]; we were able to pinpoint the precise nature of these mediatory thought processes as ruminative, not reflective. It may be that unsought and negatively focused thoughts about an event [e.g., 32] drive identity fusion because they compel the individual to fixate on the feeling of distress that was originally cast by the experience. This preoccupation with the event can then give rise to the perception that there are others in the world who have endured a life-altering experience similar to their own, thus mentally demarcating those who have shared the experience from those who have not shared the experience.

We also found that postpartum mothers who experienced less childbirth dysphoria (by way of pain levels) engaged in less ruminative thought, and postpartum mothers who engaged in less ruminative thought fused more strongly with mothers who have had an easy birth. As alluded to earlier, a deficiency in pain is linked to greater identity fusion, this time via an absence of ruminative thought.

## Is identity fusion associated with psychological health benefits?

Our fourth hypothesis was partially supported: for postpartum mothers, identity fusion was a moderator of PTG, but not of depression or PTSD. Postpartum mothers who fused more strongly with all mothers experienced greater PTG when ruminating more after childbirth, and postpartum mothers who fused more strongly with mothers who had a difficult birth experienced greater PTG when reflecting more after childbirth. These exploratory analyses

reveal that when fusion with other mothers is strong, PTG can be bolstered after childbirth via childbirth-related thought.

It may be that identity fusion has the potential to maximize the relationship between event-related thoughts and PTG, allowing for more "adversarial growth" that drives an individual towards an improved level of functioning in life [56]. Growth occurs when an individual comes to terms with a revised version of reality in the aftermath of trauma; in order to make sense of this altered reality, they have to cognitively rebuild their understanding of the world. Survivors of trauma try to draw out value and meaning from their painful experiences. As such, growth is an individual's shot at psychological survival when they realize that previous goals, philosophies, or beliefs are no longer possible, true, or achievable [57]. Perhaps the shared experience pathway to identity fusion provides conditions that are optimal for the promotion of growth. Mothers who are strongly fused with other mothers do not feel as isolated or alone after childbirth, and they are able to find solace and motivation knowing that other mothers have gone through a similar juncture in life. They are emboldened to better adapt to life after childbirth and take notice of the highlights of raising a newborn (i.e., finding a new purpose in life, bonding with the child) [e.g., 58].

Theorists propose three unique pathways to PTG: strength through suffering, psychological preparedness, and existential reevaluation. Although these pathways are not mutually exclusive, they are distinct psychological processes. Strength through suffering occurs when individuals acknowledge that they have come out of an agonizing experience with previously uncharted personal strengths and newfound skills and opportunities; psychological preparedness occurs when individuals successfully cope with a distressing experience and are compelled to accept their vulnerabilities and incorporate the tragedy into their internal world, building themselves to be psychologically safeguarded against future life tragedies; existential reevaluation occurs when individuals are confronted with adversity and are driven towards a sort of re-awakening where they have a newfound gratitude for their life, their relationships with others, and their spiritual beliefs [59]. It may be that these psychological processes are accelerated for highly fused mothers. Because they belong to a symbolic coalition of mothers, they are more inclined to recognize their own resiliency, reinforce themselves against ensuing troubles, and discover a new outlook on life.

### Limitations and future research

The current study was cross-sectional; we cannot claim causality because all links were concurrent associations. Future longitudinal studies could follow the course of identity fusion as well as event-related mechanisms of fusion over the antenatal and postpartum periods. It would also be interesting to measure other mechanisms of pain perception and epidural use [e.g., 60], endogenous oxytocin levels and exposure to synthetic oxytocin such as Pitocin [e.g., 61,62], and stress hormone levels such as cortisol [e.g., 63] at different time points before, during, and after childbirth. Our sample is not representative of first-time mothers outside the United States; future analyses could explore possible cultural differences across Western and non-Western cultures.

It is important to note that our sample did not include a sizeable proportion of women who had experienced a traumatic childbirth. Only 9.3% (7 out of 75 postpartum mothers) of our sample demonstrated a probable diagnosis of PTSD at a cutoff score of $\geq 28$ [35]. Although this ratio is well above the prevalence rate of postpartum PTSD found in general community samples [see 2], it is comparably lower than rates demonstrated in online samples (i.e., a self-selection bias) [e.g., 64,65]. This difference could be because the current study recruited first-time mothers via Qualtrics Panel Services in addition to parenting and childbirth-related

websites. Because most childbirth experiences are not perceived as traumatic, we recommend that future research compares the trajectories of first-time mothers who report experiencing a traumatic childbirth with those of first-time mothers who do not. Prospective analyses could also explore possible positive mechanisms of fusion, such as the degree of joy and happiness felt by first-time mothers after childbirth. We also recommend that childbirth narratives from first-time mothers are analyzed in order to test the contribution of autobiographical reasoning [66] and redemption themes [see 67–69] to identity fusion, event-related mechanisms of identity fusion, and PTG.

## Conclusions

This study provides a novel perspective on the social psychology of a population that has been particularly understudied in identity fusion research—women. Identity fusion is strengthened in women after experiencing an event that can sometimes feel like an isolating and physically and emotionally demanding experience—the birth of their first child. Manifestation of social bonds in the form of identity fusion brings to the fore how women can feel a sense of solidarity and camaraderie with other women who have experienced a similar event with varying degrees of intensity and overlap. The potential for identity fusion to be harnessed for psychological health gains as well as for prosocial action, after following shared life-altering experiences, should be further investigated.

## Supporting information

**S1 Appendix. Study advertisement posted on websites.**
(DOCX)

**S2 Appendix. Screening question for the antenatal and postpartum questionnaires.**
(DOCX)

**S3 Appendix. Adapted version of the Edinburgh Postnatal Depression Scale for the antenatal questionnaire.**
(DOCX)

**S4 Appendix. Second screening question for the postpartum questionnaire.**
(DOCX)

**S5 Appendix. Adapted version of the Posttraumatic Diagnostic Scale—Self-Report Version for DSM-5 for the postpartum questionnaire.**
(DOCX)

**S6 Appendix. Adapted version of the Posttraumatic Growth Inventory for the postpartum questionnaire.**
(DOCX)

**S7 Appendix. Adapted version of the Event Related Rumination Inventory for the postpartum questionnaire.**
(DOCX)

**S8 Appendix. Adapted version of the Centrality of Event Scale for the postpartum questionnaire.**
(DOCX)

## Acknowledgments

We thank all of the mothers who participated in this study. We thank the doctoral viva examiners for their comments on this research.

## Author Contributions

**Conceptualization:** Tara Tasuji, Elaine Reese, Valerie van Mulukom, Harvey Whitehouse.

**Formal analysis:** Tara Tasuji, Elaine Reese.

**Funding acquisition:** Harvey Whitehouse.

**Investigation:** Tara Tasuji.

**Methodology:** Tara Tasuji, Elaine Reese, Valerie van Mulukom, Harvey Whitehouse.

**Project administration:** Tara Tasuji, Elaine Reese, Harvey Whitehouse.

**Supervision:** Elaine Reese, Harvey Whitehouse.

**Visualization:** Tara Tasuji.

**Writing – original draft:** Tara Tasuji.

**Writing – review & editing:** Tara Tasuji, Elaine Reese, Valerie van Mulukom, Harvey Whitehouse.

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
