## [Decision Letter · Decision Letter 0]

15 Jul 2020

PONE-D-20-08734

Band of mothers: Childbirth as a female bonding experience

PLOS ONE

Dear Dr. Tasuji,

Thank you for submitting your manuscript to PLOS ONE. As previously conveyed to you, considerable challenges have been encountered securing expert reviews of this work. Comment has now been received from a leader in the field and these are appended here. Rather than prolong matters further waiting for an additional review I have decided to move forward based on the feedback already provided. As you can see, the view is that there is much work to be done to bring the study to a publishable standard. The comments are clear and so I will not restate them here, but needless to say each deserve attention. I would like to offer you the opportunity to invite you to submit a revised version of the manuscript that addresses the points raised. In the spirit of transparency, I intend to send any revised manuscript back to the reviewer for further comment.

A rebuttal letter that responds to each point raised by the reviewer. You should upload this letter as a separate file labeled 'Response to Reviewer'.A marked-up copy of your manuscript that highlights changes made to the original version. You should upload this as a separate file labeled 'Revised Manuscript with Track Changes'.An unmarked version of your revised paper without tracked changes. You should upload this as a separate file labeled 'Manuscript'.

We look forward to receiving your revised manuscript.

Kind regards,

Mark Nielsen, Ph.D.

Academic Editor

PLOS ONE

Journal Requirements:

2. Please change "Caucasian” to “White” or “of [Western] European descent” (as appropriate).

3. In your Methods section, please include additional information about your dataset and ensure that you have included a statement specifying whether the collection method complied with the terms and conditions for the websites from which you have collected data.

4. Please provide additional information about the participant recruitment method and the demographic details of your participants. The methods section lacks:

a) the recruitment date range (month and year),

b) a description of any inclusion/exclusion criteria that were applied to participant recruitment, and

c) a statement as to whether your sample can be considered representative of a larger population.

Reviewers' comments:

Reviewer's Responses to Questions

**Comments to the Author**

1. Is the manuscript technically sound, and do the data support the conclusions?

Reviewer #1: No

2. Has the statistical analysis been performed appropriately and rigorously? 

Reviewer #1: N/A

3. Have the authors made all data underlying the findings in their manuscript fully available?

Reviewer #1: Yes

4. Is the manuscript presented in an intelligible fashion and written in standard English?

Reviewer #1: No

5. Review Comments to the Author

Reviewer #1: Band of mothers: Childbirth as a female bonding experience

The study is on a topic of broad relevance. However, there are a number of concerns:

- Identity fusion is a subject of interest, but the way this was explored in this study was not convincing.

- At 46 pages, with 5 appendices in addition, 5 figures and 8 tables, the paper is difficult to read and overly long. There is considerable repetition, particularly in describing the hypotheses and research questions initially and then in the findings and the discussion.

The variation in time since giving birth (1-28 weeks) or the duration of pregnancy (1-40 weeks) at the time of survey completion is not satisfactory. We know that maternal mood and for example, symptoms of depression (as measured by the EPDS) change over the course of the postnatal period and that anxiety and depression symptoms may change over the course of pregnancy. Thus combining all the data from 1-28 weeks after the event of interest is not satisfactory.

- The assumptions made in the process of setting up and carrying out this study overemphasise the negative in framing the possible experience of childbirth in terms of trauma and post-traumatic stress disorder (PTSD). This over-pathologising the experience of childbirth and the reliance on the limited quality of small scale studies in the area of PTSD associated with childbirth, often with specific site-based data collection that relies on women requiring and receiving care at specialist centres or else from self-selected online participants, appears to have led to research questions that relate to a relatively small group (it is estimated 3%).

- The direct linking to participants of childbirth with trauma is of concern throughout the study write up and in the way it was presented to participants.

- Use of the term ‘dysphoria’ – ‘a profound state of unease or dissatisfaction’ which might affect them was not really investigated here, nor was trauma. In questions just devised for this particular study, ie not previously validated, the subject was the childbirth experience and women were simply were asked about how painful and how difficult they had expected their childbirth to be and then about pain intensity and how unpleasant the experience had been generally and then at the peak of pain. These were then combined in an average pain score, again not validated.

- The details of the measures used are presented separately in multiple appendices and there is no overall information about what women were told initially about the study that might encourage them to complete the survey or not and the basic instructions used in the online version. How the study was framed to participants is key in analyzing the data and interpreting the findings. See below:

e.g. S2 Appendix. Adapted Version of the Posttraumatic Diagnostic Scale – Self-Report Version for DSM-5 for the Postpartum Questionnaire.

Below is a list of problems that people sometimes have after experiencing a traumatic event.

Please read each statement carefully and choose the number that best describes how often that problem has been happening and how much it upset you over THE LAST MONTH. Rate each problem with respect to your childbirth experience.

- Multiple measures were used, giving rise to multiple testing in this cross-sectional study of a small population (n=75) of self-selected postnatal women who were the main target of the analyses presented. Justification of the measures selected is inadequate.

- The findings reflect associations and no causal pathways should be implied in the language used or the way the data are presented. A longitudinal population based study with appropriate measures would be needed with a considerably larger sample size.

- The correlations presented are not at all surprising. It looks as though some of these occur because what was being measured was the in effect the same thing or at least closely related, with overlaps between the contructs and co-morbidities.

- It is of concern that the measures validated and published by other authors and then selected for the study described were used with adapted wording and thus were in effect changed.

-The hypothesized links with postnatal health were not really investigated and it was not possible to do so.

In summary the paper reads like an exploratory pilot study. To be acceptable considerable editing would be required, however, there are methodological issues that it may well not be possible to address

The title ‘Band of mothers: Childbirth as a female bonding experience’ is an attractive one, but the paper and study methods do not adequately reflect this in the substantial material offered.

The contributions and role of the author and co-authors of are not provided and should be clear.

6. PLOS authors have the option to publish the peer review history of their article (what does this mean?). If published, this will include your full peer review and any attached files.

Reviewer #1: No

---

## [Author Response · Author response to Decision Letter 0]

28 Aug 2020

August 28, 2020

Dear Dr. Nielsen:

Thank you for your comments on our paper and those of the reviewer. Please consider the revised version of our research article “Band of mothers: Childbirth as a female bonding experience” for publication in PLoS ONE. We have addressed each point raised by the reviewer.

The study is on a topic of broad relevance. However, there are a number of concerns:

- Identity fusion is a subject of interest, but the way this was explored in this study was not convincing.

We have clarified our specific reasons for why identity fusion in mothers is interesting and important. Please see pp. 3-5 in the introduction and 34-35 in the discussion.

- At 46 pages, with 5 appendices in addition, 5 figures and 8 tables, the paper is difficult to read and overly long. There is considerable repetition, particularly in describing the hypotheses and research questions initially and then in the findings and the discussion.

We have radically streamlined the manuscript to cut out 11 pages from our original submission. In the process, we have rewritten and streamlined the hypotheses so that they are now clearer.

The variation in time since giving birth (1-28 weeks) or the duration of pregnancy (1-40 weeks) at the time of survey completion is not satisfactory. We know that maternal mood and for example, symptoms of depression (as measured by the EPDS) change over the course of the postnatal period and that anxiety and depression symptoms may change over the course of pregnancy. Thus combining all the data from 1-28 weeks after the event of interest is not satisfactory.

The weeks of pregnancy and age of baby variables were not significantly correlated with identity fusion. For the antenatal group, the weeks of pregnancy variable was also not significantly correlated with expected pain and difficulty. For the postpartum group, the age of baby variable was also not significantly correlated with perceived sharedness, pain, and the thinking measures. We have presented this information in the preliminary analyses section on pp. 20.

- The assumptions made in the process of setting up and carrying out this study overemphasise the negative in framing the possible experience of childbirth in terms of trauma and post-traumatic stress disorder (PTSD). This over-pathologising the experience of childbirth and the reliance on the limited quality of small scale studies in the area of PTSD associated with childbirth, often with specific site-based data collection that relies on women requiring and receiving care at specialist centres or else from self-selected online participants, appears to have led to research questions that relate to a relatively small group (it is estimated 3%).

We have reframed the introduction to note that childbirth is not a traumatic event for most women; indeed, it is typically a joyful and life-changing event. However, it is a physically demanding and highly emotional experience for all women, so it is a good candidate for fusion. The prevailing theory when we began this research was that highly dysphoric events led to fusion. Since that time, there are suggestions that intensely positive events can also lead to fusion. We have cited this research in the introduction, and noted as a limitation in the discussion that we did not measure childbirth-related joy, which we suggest would be useful in future research. Because our study is the first to test childbirth-related fusion, we hope that other researchers will be inspired to refine and expand upon our initial findings in future studies.

- The direct linking to participants of childbirth with trauma is of concern throughout the study write up and in the way it was presented to participants.

We have reframed the introduction and the discussion to focus less on trauma and more on the emotional intensity of childbirth. In terms of our presentation of the study to participants, we do not believe that it would have caused harm to participants, and our instructions were thoroughly screened by the Ethics Committee at the University of Oxford. 

- Use of the term ‘dysphoria’ – ‘a profound state of unease or dissatisfaction’ which might affect them was not really investigated here, nor was trauma. In questions just devised for this particular study, ie not previously validated, the subject was the childbirth experience and women were simply were asked about how painful and how difficult they had expected their childbirth to be and then about pain intensity and how unpleasant the experience had been generally and then at the peak of pain. These were then combined in an average pain score, again not validated.

It is true that some of these are new measures developed specifically for our novel purposes. The measures all displayed adequate reliability as detailed on pp. 16. Thus, we believe they are promising, although we acknowledge they will require further validation in future studies.

- The details of the measures used are presented separately in multiple appendices and there is no overall information about what women were told initially about the study that might encourage them to complete the survey or not and the basic instructions used in the online version. How the study was framed to participants is key in analyzing the data and interpreting the findings. See below:

e.g. S2 Appendix. Adapted Version of the Posttraumatic Diagnostic Scale – Self-Report Version for DSM-5 for the Postpartum Questionnaire.

Below is a list of problems that people sometimes have after experiencing a traumatic event.

Please read each statement carefully and choose the number that best describes how often that problem has been happening and how much it upset you over THE LAST MONTH. Rate each problem with respect to your childbirth experience.

We have clarified these issues. Please see pp. 10 in the text and S1 Appendix for the study advertisement that was posted on all parenting and childbirth-related websites.

- Multiple measures were used, giving rise to multiple testing in this cross-sectional study of a small population (n=75) of self-selected postnatal women who were the main target of the analyses presented. Justification of the measures selected is inadequate.

It is commonplace to use multiple measures in an exploratory study such as this one, which is the first to address fusion in new mothers. Our streamlined hypotheses have clarified that we are looking for patterns across these measures. 

- The findings reflect associations and no causal pathways should be implied in the language used or the way the data are presented. A longitudinal population based study with appropriate measures would be needed with a considerably larger sample size.

We have deleted all instances of causal language.

- The correlations presented are not at all surprising. It looks as though some of these occur because what was being measured was the in effect the same thing or at least closely related, with overlaps between the contructs and co-morbidities.

With our streamlined hypotheses, we have cut down on the number of correlations presented.

- It is of concern that the measures validated and published by other authors and then selected for the study described were used with adapted wording and thus were in effect changed.

As previously indicated, we needed new measures for this novel study; our adaptations will need to be further validated in future research.

-The hypothesized links with postnatal health were not really investigated and it was not possible to do so.

We acknowledge in the discussion section on pp. 35 that these links are concurrent associations, so we cannot claim causality here.

In summary the paper reads like an exploratory pilot study. To be acceptable considerable editing would be required, however, there are methodological issues that it may well not be possible to address

Our study is the first to research the phenomenon of childbirth-related fusion, so in that sense it is exploratory. However, with over 160 participants and two groups of mothers (antenatal and postpartum), it went well beyond being a pilot study, which typically would be with 10-20 participants. With the reduction in pages and improved use of language, we hope that the paper now better communicates its significance.

The title ‘Band of mothers: Childbirth as a female bonding experience’ is an attractive one, but the paper and study methods do not adequately reflect this in the substantial material offered.

We believe that this title accurately reflects our main finding, which was that postpartum mothers demonstrated higher levels of fusion with other mothers compared to antenatal mothers. Thus, the event of childbirth supports bonding of women with other mothers.

The contributions and role of the author and co-authors of are not provided and should be clear.

We have provided this information in the submission system. We have also provided this information here for your reference: 

Tara Tasuji: Conceptualization, formal analysis, investigation, methodology, project administration, visualization, writing – original draft preparation, writing – review & editing.

Elaine Reese: Conceptualization, formal analysis, methodology, project administration, supervision, writing – review & editing.

Valerie van Mulukom: Conceptualization, methodology, writing – review & editing.

Harvey Whitehouse: Conceptualization, funding acquisition, methodology, project administration, supervision, writing – review & editing.

Sincerely,

Tara Tasuji

---

## [Editor Report · Decision Letter 1]

22 Sep 2020

Band of mothers: Childbirth as a female bonding experience

PONE-D-20-08734R1

Dear Dr. Tasuji,

We’re pleased to inform you that your manuscript has been judged scientifically suitable for publication and will be formally accepted for publication once it meets all outstanding technical requirements.

Kind regards,

Mark Nielsen, Ph.D.

Academic Editor

PLOS ONE
---

## [Editor Report · Acceptance letter]

30 Sep 2020

PONE-D-20-08734R1 

Band of mothers: Childbirth as a female bonding experience 

Dear Dr. Tasuji:

I'm pleased to inform you that your manuscript has been deemed suitable for publication in PLOS ONE. Congratulations! Your manuscript is now with our production department. 

Kind regards, 

on behalf of

Dr. Mark Nielsen 

Academic Editor

PLOS ONE